# Are LLM Agents Behaviorally Coherent? Latent Profiles for Social Simulation

## Abstract

The impressive capabilities of Large Language Models (LLMs) have fueled the notion that synthetic agents can serve as substitutes for real participants in human-subject research. To evaluate this claim, prior research has largely focused on whether LLM-generated survey responses align with those produced by human respondents whom the LLMs are prompted to represent. In contrast, we address a more fundamental question: Do agents maintain *internal consistency*, retaining similar behaviors when examined under different experimental settings? To this end, we develop a study designed to (a) reveal the agent's internal state and (b) examine agent behavior in a conversational setting. This design enables us to explore a set of behavioral hypotheses to assess whether an agent's conversational behavior is consistent with what we would expect from its revealed internal state. Our findings show significant internal inconsistencies in LLMs across model families and at differing model sizes. Most importantly, we find that, although agents may generate responses matching those of their human counterparts, they fail to be internally consistent, representing a critical gap in their capabilities to accurately substitute for real participants in human-subject research.

## 1 Introduction

LLMs have demonstrated remarkable progress in recent years, prompting researchers and practitioners alike to ask not whether these systems can pass the Turing test Jones & Bergen (2025), but whether they can convincingly adopt full-fledged human personas Hu & Collier (2024); Park et al. (2023). Early findings suggest they can. For example, Park et al. (2024) finds that when agents are constructed using rich qualitative interview data, they exhibit attitudes and behaviors that closely mirror those of their human counterparts. Such results have inspired what we term the *substitution thesis*: if agents can emulate humans, they may serve as substitutes for real participants in human-centered research. As substitutes, agents can be examined for individual traits or can be deployed to simulate human societies at scale. Should this prove viable, the potential upsides for social research would be tremendous: companies might test new products on virtual customers (Xiang et al., 2024; Ilagan et al., 2024) , and social scientists could explore complex phenomena like war Hua et al. (2024), governance Piatti et al. (2024), or cultural evolution Perez et al. (2024) with fewer ethical and logistical constraints.

Still, the gap between technological promise and practical utility remains large. While Park et al. (2024) achieves impressive persona fidelity, it does so by relying on lengthy two-hour interviews. Indeed, in studies where agents are not given such extensive background information, their persona mimicry begins drifting in significant ways. For instance, when tasked with representing different American sociopolitical groups, LLM agents broadly matched aggregate human opinions but displayed far less variance, raising doubts about their use in downstream analyses (Bisbee et al., 2024). Similar "flattening effects" have also been observed across identity groups, where agent responses appear more homogeneous than their real-world counterparts (Wang et al., 2025a). We make further strides in identifying current gaps in the substitution thesis with the following contributions.

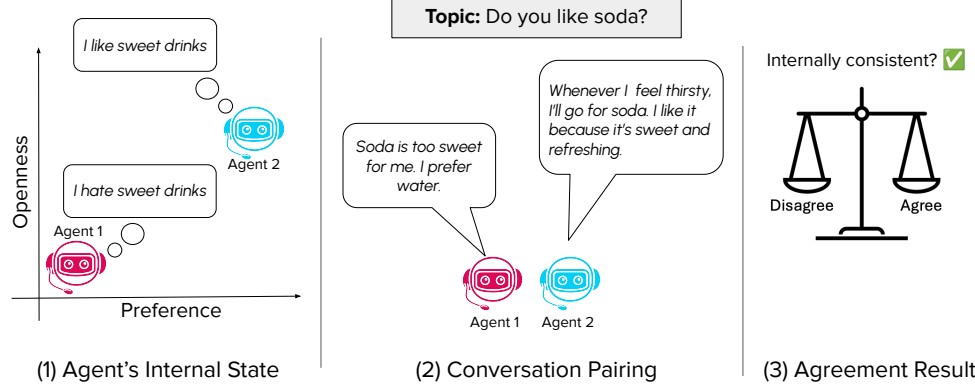

Figure 1: A proposed framework for probing behavioral consistency in LLM agents. Rather than comparing LLM-predicted latent profiles against their human counterparts, we compare the predicted LLM agents' latent profiles against each other based on ground-truth models. Each agent is assigned a latent profile consisting both their openness to change their mind and their stance on a specific topic (e.g., a preference for sweet drinks). We then measure if the agreement level of the conversation is consistent with the latent profiles of the pairs. For instance, Agent 1 is internally consistent with its latent profile ("dislike sweet drinks", "low openness score") since at the end of the conversation, it still prefers water.

**Proposed Framework**. Our design rests on two pillars: (1) inferring an agent's internal state, and (2) testing whether it manifests in conversation with others, as stated in Figure 1. For internal state, we solicit both the agent's preference on a topic and its openness to persuasion. One agent may strongly dislike taxes, while another supports them; one may admit to being easily swayed, while another identifies as stubborn. We leverage this heterogeneity in the agents' preferences and openness for our second step, the conversation pairing. If agents are consistent with their internal state, then we ought to be able to manufacture agreement or disagreement based on who we pair in a conversation. Simply put, those with aligned views should agree more than those with opposed views. Moreover, the flexibility in pairing agents across topic, preference, and openness ensures that our design guards against a well-known issue: humans themselves are not always consistent in acting on their preferences. Our study addresses this problem by shifting from individual reliability to collective patterns. By comparing how different agent pairings interact, we capture group-level consistency (or lack thereof) and bypass the need for any single agent's self-report to be fully accurate. This approach reduces noise from the individual level and reveals the more stable signals that emerge across groups.

**Findings**. Pairs with aligned sentiments often reach the highest levels of agreement, yet the reverse is not true – agents with opposing preferences rarely disagree outright. Instead, their conversations typically end in neutral outcomes, making actual disagreement a notable exception in our study. This pattern holds even when accounting for sycophancy as usually proscribed by other researchers (Sharma et al., 2025). Furthermore, even among agents who share the same sentiment, we observe meaningful differences based on whether that sentiment is positive or negative. For example, a mutual dislike of a topic such as taxes should, in theory, lead to the same level of agreement as mutual appreciation. However, our results show that shared dislike consistently yields lower agreement.

## 2 Related Work

### 2.1 LLMs as Agents and Human Substitutes in Dialogue

Recent advances in Large Language Models (LLMs) have opened new possibilities for simulating human subjects in social science research. These models exhibit context-sensitive reasoning and structured decision-making capabilities (Wei et al., 2022; Kojima et al., 2022), enabling researchers to utilize them not only as tools but as experimental subjects (Mou

et al., 2024; Park et al., 2023). In multi-agent simulations, LLMs have demonstrated socially emergent behaviors—forming memories, goals, and interaction patterns resembling real-world dynamics (Wang et al., 2025b). They have been used to model phenomena like conformity, information cocoons (Anthis et al., 2025), war (Hua et al., 2024), and market competition (Zhao et al., 2024). In structured survey settings, their responses have shown high alignment with human data across various conditions (Anthis et al., 2025). Nonetheless, significant conceptual and technical challenges remain. LLMs rely on statistical prediction rather than cognitive reasoning, and while they may appear behaviorally plausible, this can obscure underlying instability. They often fail to reproduce human-like distributional variance or demographic nuance and remain highly sensitive to prompt design and temporal drift (Bisbee et al., 2024; Petrov et al., 2024; Takata et al., 2024).

## 2.2 BEHAVIORAL CONSISTENCY AMONG PERSONALITY, PREFERENCE, AND TOPIC

Although LLMs can maintain fluent conversation, they frequently lack continuity in personality and preference across multiple turns. Benchmarks like Topic-Conversation Relevance (TCR) assess topic relevance (Fan et al., 2024), but do not account for how personality traits might influence topic engagement or behavioral adaptation. Similarly, Long-Term Memory (LTM) benchmarks show that while LLMs can recall factual details, they struggle to retain identity- or preference-linked information over time (Castillo-Bolado et al., 2024).

Traditional persona-based models Zhang et al. (2018); Rashkin et al. (2019) allow for stylized variation (e.g., "likes cats"), but do not simulate evolving personality states or trait-informed reasoning. Recent works on generative agents with memory and reflection (Park et al., 2023) and trust-aware simulations (Xie et al., 2024) have made progress toward this goal, yet fall short in capturing how personality shapes topic alignment in dynamic conversations. Persona injection has been shown to improve coherence and emotional nuance (Wu et al., 2025). Trait-grounded personas help LLMs maintain consistent behaviors, influencing both the form and distribution of emotional support strategies. Synthetic datasets built from large-scale simulations further show that persona conditioning enhances diversity across psychological traits (Ge et al., 2024; Wu et al., 2025). However, challenges still remain, as studies have shown that dialogues generated without personas tend to be more concentrated and less diverse in psychological traits. In contrast, persona-conditioned outputs distribute more broadly across trait dimensions, such as Emotionality and Openness (Wu et al., 2025). The Big Five traits, including openness, are both stable across time and life events (Cobb-Clark & Schurer, 2012) and significantly correlated with resilience, cognitive flexibility, and adaptive functioning (Oshio et al., 2018).

## 3 A FRAMEWORK FOR PROBING BEHAVIORAL COHERENCE

### 3.1 EXPERIMENTAL FRAMEWORK

At a high level, our framework's tests undergo five sequential stages as shown in Figure 2: (1) select a topic, (2) generate agents, (3) elicit their internal states, (4) pair them for dialogue, and (5) evaluate conversational agreement (details in Appendices A and B).

**Topic Selection:** Construct a set of topics, where each topic $T$ is associated with a contentiousness level $C \in \{1, 2, 3\}$, with 1 being the least contentious and 3 being the most contentious . The set contains nine topics in total, with exactly three topics assigned to each contentiousness level. Further descriptions of the topics may be found in Table 3.

**Generate Agents:** For each topic, construct agents with demographic profiles $D_i$ defined by age, gender, urbanicity, location, and education (see Appendix B for more details on specific prompt construction). We further modify the agent prompt to include their bias towards the topic at hand, $B \in \{1, 2, 3\}$, with 1 being the least biased towards an opinion on a topic, and 3 being the most biased. Demographic region is limited to the United States and systematically varied across 5 age groups, 2 genders, 4 regions, 4 urbanicity levels, and 6 education levels.

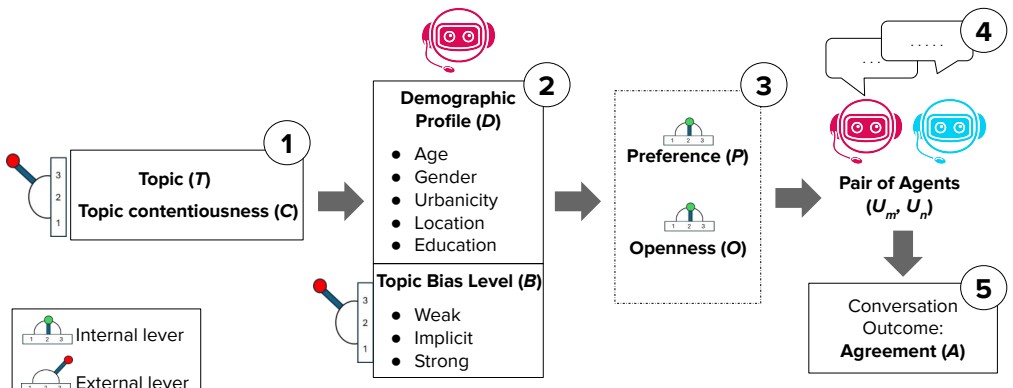

Figure 2: Proposed framework for probing behavioral coherence : (1) We first gather a set of topics of varying contentiousness levels to query agents on. (2) For a given topic, explicit agent profiles are gathered by varying the prompt among different demographic values (age, gender, etc.). This prompt is further altered to include information specific to the agent's bias toward the chosen topic. (3) Internal states are gathered for each agent by asking a question about their preference on the given topic and about their openness to being swayed by others. (4) Agents are paired together to discuss the topic, and (5) agreement scores are calculated for each turn of their conversations.

**Internal State:** Identify each agent's preference $P_i$ and openness $O_i$. Both these values are captured by posing a set of questions to agent $i$ and the responses are used to create a number indicating an agent's preference for a topic and their openness. Section A details how these values are determined in greater detail.

**Pairing Agents:** Each agent $i$ has a profile $(P_i, O_i, B_i)$. Let $U := \{(P_j, O_j, B_j)\} \, \forall j \in \{1, .., N\}$ be the set of all $(P, O, B)$ tuples (representing all possible agents in our setup). For all possible pairs $(U_m, U_n)$, we sample agents $q, r$ such that $(P_q, O_q, B_q) = U_m$ and $(P_r, O_r, B_r) = U_n$.

**Conversation Outcome:** For each step $K$ of a conversation, we use LLM-as-judge to score agreement $A \in \{1, 2, 3, 4, 5\}$ (1 = complete disagreement, 5 = complete agreement). For analysis, we retain only the final agreement score. To calibrate judgments, we provide five annotated sample conversations—one for each score.

## 3.2 SPECIFIC EXPERIMENTS

We examine 9 topics with different controversy levels to explore the effect of contentiousness on both individual agent response as well as on agreement during interaction with other agents. More details on specific topics and agent demographics can be found in Appendix A (in particular, Tables 3 and 4 show the set of topics and agent demographics we consider).

Using these topics and demographics, we conduct two sets of experiments: (1) qualitative experiments (§4.1 and §4.2) as a proof-of-concept on LLMs' internal consistency on preferences and openness, and (2) formalized experiments (§4.3) to compare various LLMs' performances for robustness. For both experiment sets, we use `Llama3.1-8B-Instruct` (et al., 2024) as the judge (except in cases where we evaluate `Llama3.1-8B-Instruct` as the agent model, in which case we use `Qwen2.5-7B-Instruct` (Qwen et al., 2025) as the judge). For the qualitative experiments below, we use `Qwen2.5-7B-Instruct` as the agent model. For the formalized and aggregated experiments, we examine `Qwen2.5` models with various sizes (3B, 7B, and 14B) as well as other model types (`Llama3.1-8B`, `Gemma-2-9B` (Team, 2024), `Mistral-7B` (Jiang et al., 2023), and `Olmo-7B` (Groeneveld et al., 2024)) as the agents.

## 4 EXPERIMENTAL RESULTS

In our experiments, we aim to uncover the discrepancy between the *appearance* of behavioral consistency and its breakdown under closer examination. To this end, we organize our findings along two dimensions – **preference** and **openness** – using bootstrap sampling as our primary mode of analysis. For more details, refer to Appendix C.

### 4.1 ON PREFERENCES

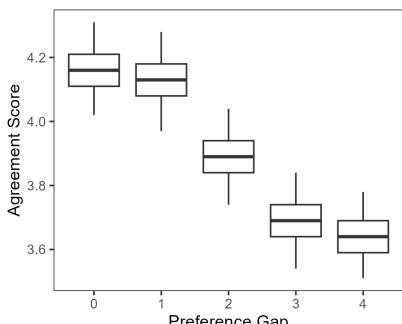

Figure 3: Average agreement score when conversation pairs are grouped according to their preference gap

**Finding #1: Increasing Preference Gap Decreases Agreement** The most basic consistency test is whether agents with shared preferences are more likely to agree than those with divergent ones. We examine this by computing the *preference gap* as the absolute difference in two agents' preference scores ($|1-5|$). A gap of four indicates maximal difference whereas a gap of zero indicates identical views. 3 displays our results, which appear consistent with our expectation: Pairs with aligned preferences (gap=0) achieve the highest agreement, while pairs with greater gaps yield progressively lower scores. Agreement, thus, decreases with larger preference gaps.

Stopping the analysis here would misleadingly suggest that agents are behaviorally consistent. In fact, the results in Figure 3 conceal significant inconsistencies. A deeper analysis reveals three systematic discrepancies: (1) disagreement is strongly dampened between agents with dissimilar preferences, (2) shared negative sentiment produces weaker alignment than shared positive sentiment, and (3) topic contentiousness exerts undue influence on outcomes.

**Finding #2: Agreement amplifies, disagreement dampens:** Figure 3 shows that agreement levels remain high even at large preference gaps. Instead of converging toward the minimum score of 1, pairs with maximal divergence average between 3.5 and 3.7 with fewer than 1% of all conversations yielding scores below 3. In other words, agents virtually never disagree, no matter how much they diverge in their preferences. One potential explanation for these patterns is sycophancy, a bias in which language models tend to be overly agreeable toward their interlocutor. In our context, such sycophancy would manifest as a systematic amplification of agreement and a corresponding suppression of disagreement.

| Agreement Score | Observed Probability Preference Gap = 0 | Expected Probability Preference Gap = 4 |
|---|---|---|
| 1 | 0.000204 | 0.371 |
| 2 | 0.00163 | 0.423 |
| 3 | 0.205 | 0.205 |
| 4 | 0.423 | 0.00163 |
| 5 | 0.371 | 0.000204 |

Table 1: To estimate how much preference is being depressed, we assume that the expected disagreement pattern for pairs with the widest preference gap is the inverse of pairs with a preference gap of zero.

To uncover the effects of sycophancy, we estimate the suppression of disagreement by adopting a simplifying assumption–agents should, in principle, disagree at the same rate as they agree. We already know one end of this spectrum, namely, the amount of agreement when agents are perfectly aligned (gap=0). We establish the other end of the spectrum, the disagreement side, by assuming it to be the inverse of agreement as shown in Table 1. This procedure yields the counterfactual outcomes shown in Figure 4.

For pairs with a preference gap of four, the observed mean agreement score (3.6) is roughly twice the expected value (1.8), revealing substantial suppression of disagreement. This

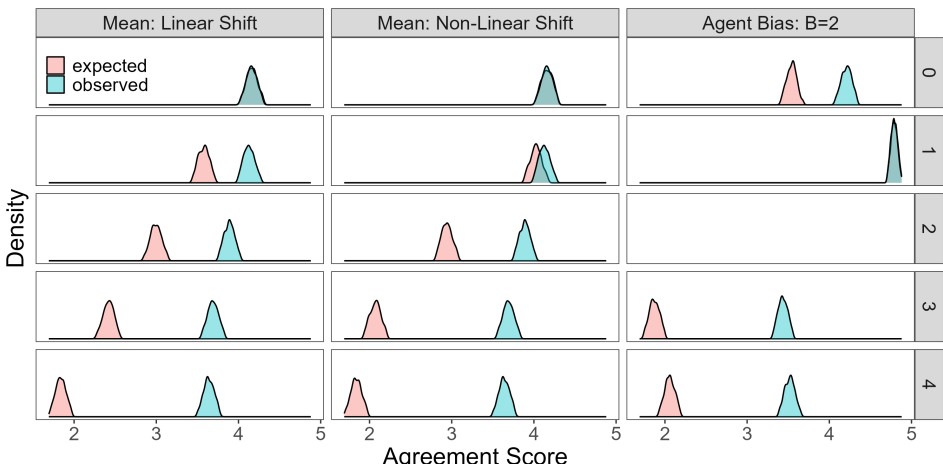

Figure 4: Estimates for disagreement suppression where the observed and expected agreement distributions are compared across preference gaps. Each row corresponds to the difference in preference on the topic of interest. The first two columns use the entire sample, but differ in the way the means shift across preference gaps. The third column is based on a subsample where the agents' topic bias is set to $B_i = 2$. The right column shows the preference gap.

effect extends across smaller gaps as well. As shown in the left column of Figure 4, when assuming the distributions shifts linearly from preference gap zero to four, the differences between expected and observed means are 1.8, 1.27, 0.89, and 0.55 for gaps four through one, corresponding to suppression magnitudes of 2.0, 1.53, 1.30, and 1.15 respectively. Replicating the analysis with a sigmoidal shift displayed in the middle column yields the same conclusion: disagreement is consistently depressed across all preference gaps. Furthermore, looking at the right column of Figure 4, this suppression persists even when agents are explicitly instructed to adopt firm positions ($B_i = 2$). Although there is an anomaly in that the highest agreement occurs at a gap of one rather than zero, the broader result is clear: disagreement is systematically dampened despite efforts to mitigate sycophancy.

**Finding #3: Shared sentiment, divergent speech destinations:** For the second hidden inconsistency, consider two pairs of agents: one strongly favors a topic, while the other strongly dislikes it. In principle, both pairs should exhibit high agreement—one through shared enthusiasm, the other through shared aversion. Sentiment alignment, whether positive or negative, should yield comparable agreement outcomes. Yet this is not what we observe.

We illuminate this inconsistency by decomposing each preference gap level into two subgroups. In the first, one agent is fixed at a preference of one; in the second, one agent is fixed at five. If agents behaved consistently, the agreement distributions of these subgroups would overlap. Instead, Figure 5 shows a systematic asymmetry. Across all cases, pairs anchored at one display lower agreement scores than those anchored at five. Moreover, this disparity grows as the preference gap narrows—from a mean difference of 0.33 at gap three to 0.62 at gap zero.

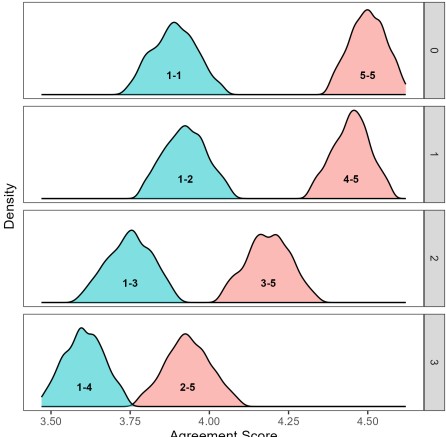

Figure 5: Agreement scores for pairs with fixed sentiment anchors. One of the agents' preference is fixed at 1 and 5 for the two groups respectively. The specific preference pairing is noted inside the distribution.

Strikingly, the (1,1) pair aligns more closely with (2,5)—three preference levels apart—than with its supposed counterpart (5,5). These results suggest that LLM agents systematically penalize expressions of strong negative sentiment.

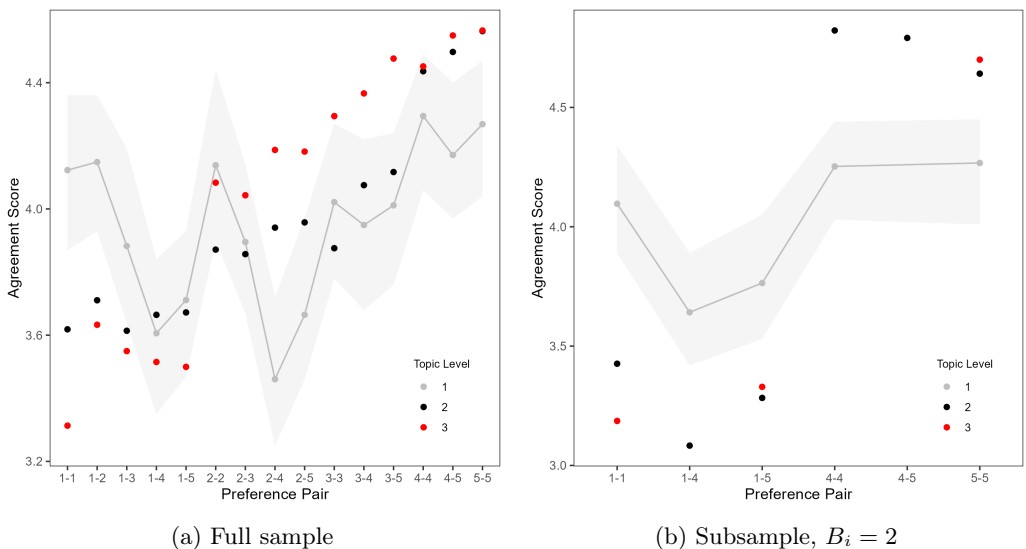

(a) Full sample                                    (b) Subsample, $B_i = 2$

Figure 6: The two graphs show whether topic controversy has an independent effect on agreement outcomes when preferences of conversation pairs are held constant. Gray bands display the confidence interval for the least controversial topics.

**Finding #4: Topic contentiousness trumps preference:** Our third indicator of inconsistency concerns the undue influence of topic contentiousness on agreement outcomes. While some subjects, such as taxes or immigration, are commonly perceived as polarizing, this reflects how individuals position themselves, not an intrinsic property of the topic. Even seemingly benign discussions, such as favorite sports or seasons, can become contentious if participants hold opposing views. Hence, pairs with the same sentiment should exhibit similar levels of agreement regardless of topic. To test this, we disaggregated agents into preference pairs (e.g., (1,3), (2,4)) and calculated agreement scores at three levels of contentiousness: neutral ($C = 1$), medium ($C = 2$), and high ($C = 3$). Using neutral topics as a baseline, we then examined how agreement shifts as topics become more controversial.

As shown in Figures 6a and 6b, agreement is strongly shaped by topic contentiousness. For the full sample, 11 of 15 preference pairs at $C = 3$ fall outside the neutral baseline's confidence interval. Scores dip below baseline for lower-ranked pairs (e.g., (1,3)) but rise above baseline for higher-ranked pairs (e.g., (3,4)). This effect grows stronger when focusing on agents instructed to take firm positions ($B_i = 2$). In Figure 6b, all six preference pairs diverge significantly from the neutral reference under medium and high contentiousness. Even with fixed preferences and reinforced stances, topic contentiousness remains the dominant factor shaping agreement outcomes - which we would not expect if preference accurately reflected the sentiment of the agent towards a topic.

## 4.2 On Openness

So far, we have focused on preferences as the primary dimension of internal state. We now turn to openness, which captures how readily an agent may be persuaded by its interlocutor. Greater openness should translate into higher levels of agreement, particularly when agents begin with divergent preferences. We observe the same pattern here by our setup which allows us to mechanically compare agents with different openness levels (as described in Section 3.1) – surface-level consistency belies deeper behavioral inconsistencies, only visible upon closer scrutiny. Figure 7a shows that as openness increases, average agreement also rises, aligning with expectations that more receptive agents converge more readily with their partners.

However, when we isolate cases of maximal preference divergence (pairs (1,5)), the pattern weakens considerably. Figure 7b compares openness pairings against the baseline (0,0), where both agents are stubborn. If openness worked as expected, all other pairings would show

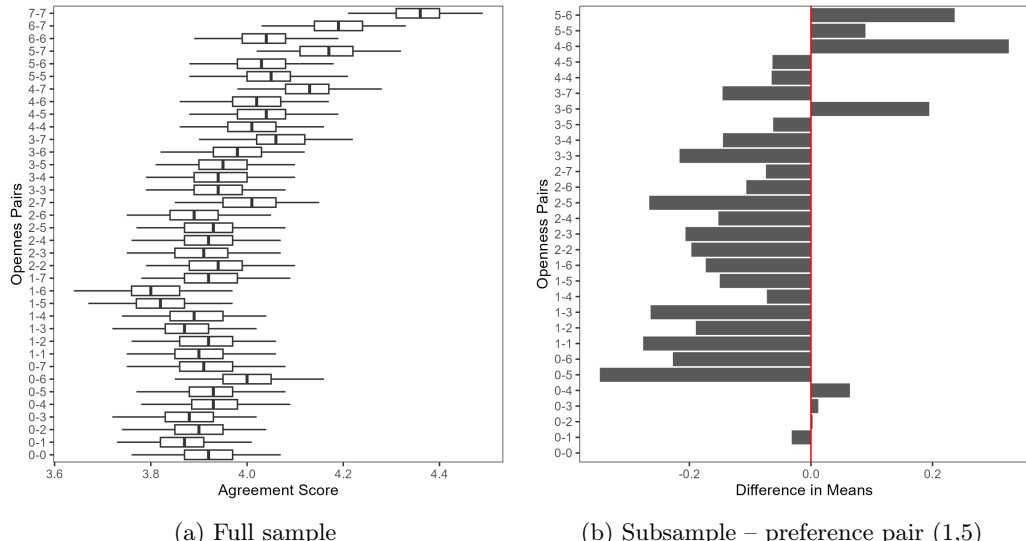

(a) Full sample        (b) Subsample – preference pair (1,5)

Figure 7: Agreement scores for pairs grouped according to how open agents are. In panel (b), the mean of all interactions is calculated for each openness pairing where the preference pair is (1, 5). Each horizontal bar represents the mean difference in agreement score between a given openness pairing and the openness pairing of (0, 0).

higher agreement than this baseline. Instead, most differences are negative, with only 6 of the 28 pairings producing greater agreement.

These findings highlight an important inconsistency. While aggregate results suggest that openness drives higher agreement, closer inspection shows this effect vanishes when agents begin from diametrically opposed positions. In such cases, openness fails to reliably increase agreement, revealing a gap between surface plausibility and deeper behavioral coherence.

## 4.3 Formalized & Aggregated Results

The qualitative findings above suggest systematic inconsistencies in how LLM agents realize their internal states. To formalize these results, we define six explicit tests of behavioral consistency. A model is considered to *pass* a test if its outcomes align with expectations of internal coherence. We use a conservative threshold of $p < .01$, applying Bonferroni corrections where multiple comparisons are made.

**Test 1: Increasing Preference Gap Decreases Agreement**. The first preference test discussed in Section 4.1. We perform a Pearson's correlation test on the agreement score and preference gap to determine if a negative relationship exists between the two.

**Test 2: Agreement Amplifies, Disagreement Dampens** We test whether the distribution of the agreement scores with preference gap of 4 has a distribution equivalent to that of the inverse of those with preference gap of 0. We test this symmetry with a Kolmogorov–Smirnov test.

**Test 3: Shared Sentiment, Divergent Speech Destinations** Pairs with identical negative preferences (1–1) should agree as much as pairs with identical positive preferences (5–5). We test whether (1–1) scores exceed those of mixed pairs (2–5, 3–5, 4–5) using Mann–Whitney U, with Bonferroni correction.

**Test 4: Topic Contentiousness Trumps Preference** We first first compute Mann-Whitney U tests independently for each preference pair and topic level pairing. For instance, we compute this test for the Preference pairing '1-1' when comparing the distributions for Topic Levels of '1' and '2'. We repeat this for all possible Preference pairings and Topic Level pairings to see if they are from the same distribution. We then aggregate these using the Bonferroni correction.

| | | Preference | | | | Openness | |
|---|---|---|---|---|---|---|---|
| | | Surface-level | In-depth | | | Surface-level | In-depth |
| | | Test 1 | Test 2 | Test 3 | Test 4 | Test 5 | Test 6 |
| Sizes | Qwen2.5-3B | ✓ | × | × | × | ✓ | × |
| | Qwen2.5-7B | ✓ | × | × | × | ✓ | × |
| | Qwen2.5-14B | ✓ | × | × | ✓ | ✓ | × |
| Types | Llama3.1-8B | ✓ | × | × | × | ✓ | ✓ |
| | Gemma-9B | ✓ | × | × | ✓ | ✓ | × |
| | Mistral-7B | ✓ | × | × | × | ✓ | × |
| | Olmo-7B | ✓ | × | × | × | ✓ | ✓ |

Table 2: Significance testing results for each model across six tests. A (✓) indicates the model passed the test, while (×) indicates failure. In all cases above, we use Llama3.1-8B-Instruct as the judge model, except when Llama3.1-8B-Instruct is being evaluated (in which case, we use Qwen2.5-7B-Instruct as the judge in order to reduce the known effects of LLM-as-judge bias towards one's own outputs).

**Test 5: Increasing Openness, Increasing Agreement Score** The first openness test discussed in Section 4.2. We perform Pearson's correlation test on the combined sum of each agents' openness score with the agreement score to determine if there is a positive linear relationship in-between. All preference pairings are considered in this test.

**Test 6: High Preference Gap and Low Openness, Too Strong Agreement** We test whether the agreement score distribution for the lowest Openness pairing and largest Preference pair difference '1-5' is smaller than the agreement scores for other pairings. Similar to Test 4, we compute each comparison separately, then perform a Bonferroni correction on a Mann-Whitney U test.

From the summarized results of Table 2, we see that all models exhibit success in Tests 1 and 5, demonstrating that increasing the preference gap lowers agreement, while increasing openness raises agreement, as expected. However, we find that many models exhibit similar, troubling results to those found in `Qwen2.5-7B-Instruct` above. No model succeeded in either Tests 2 or 3. There were some successful tests for 4 and 5; however, these use Bonferroni corrections over a large number of independent tests and are thus more conservative estimates. Overall, these results show that while models capture broad, surface-level trends, they systematically fail tests requiring deeper internal coherence. Importantly, this pattern holds across model sizes and families, suggesting that such inconsistencies are not idiosyncratic but general properties of current LLMs.

## 5 Conclusion and Discussion

This paper set out to examine the substitution thesis: the idea that LLM agents might serve as substitutes for humans in social and behavioral research. Our contribution has been to shift the focus from *external consistency* (i.e., alignment with human survey responses or demographic priors) to a more fundamental criterion: *internal behavioral consistency*. Specifically, we asked whether LLM agents behave in ways that are coherent with their own inferred internal states.

Our results reveal clear limitations in current LLM agents. While agents often appear consistent on the surface, closer inspection shows systematic deviations. Across settings, agents suppressed disagreement, favored positive over negative sentiment, and allowed topic contentiousness to overly shape outcomes. These patterns persisted across model families and sizes, indicating they are not artifacts of a single architecture but reflect broader limitations of current LLMs. The implications are significant for using LLMs in social simulation and behavioral modeling. Although these systems can produce human-like responses in isolated cases, they fail to sustain trait-driven coherence across contexts, raising doubts about their reliability as stand-ins for real human participants.

## Reproducibility Statement

We have taken several steps to facilitate the reproducibility of our work. A high-level description of our prompting methodology and experimental setup is provided in Section 3. Appendix A expands on the general framework and design choices underlying our approach, while Appendix B provides detailed specifications of the prompt configurations and experimental procedures used in our evaluations. Appendix C describes how final results are computed and reported using bootstrapping. Finally, Section 4 details the statistical tests performed to determine each model's internal consistency. Together, these materials should allow researchers to replicate our study and validate our findings.

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

## A    GENERIC DESIGN PRINCIPLES

The experimental design elicits internal states from agents and tests whether these states manifest consistently in dialogue. Below, we describe each component in turn.

**Topic Contentiousness ($C$)**   It is commonly accepted that some topics are inherently more polarizing than others. A discussion on taxes is likely to provoke more disagreement than a conversation on the weather. And yet, any topic has the potential to polarize once the right mix of people is involved. From obscure debates over historical events to the football fans rooting for rival teams, seemingly innocuous subjects can spark intense disagreement when divergent viewpoints collide. By varying the contentiousness level of the topic at hand, we assess whether our agents exhibit consistent behavior–disagreeing when their preferences diverge, regardless of the subject matter. We assign each topic a contentiousness score $C \in \{1, 2, 3\}$, where 1 is least contentious and 3 is most.

**Bias in Prompting ($B_i$)**   LLMs are known to exhibit sycophancy, often being overly agreeable to interlocutors. To counteract this, we introduce a bias parameter $B_i$ that explicitly directs agents to take a stance.

- $B = 1$: No bias information added.
- $B = 2$: Implicit biasing (e.g., "You are a liberal Democrat" when the topic is immigration).
- $B = 3$: Explicit biasing (e.g., "You support immigration" when the topic is immigration).

For $B = 2$ and $B = 3$, agents are further directed to adopt either a positive or negative position on the topic.

**Preference ($P_i$)**   Our primary measure of internal state is $P_i$, an agent's preference on a given topic. The expectation is straightforward: preferences should predict conversational outcomes. Agents with aligned preferences should agree, while those further apart should be more likely to disagree. To elicit $P_i$, we prompt agents to take a position on statements such as *"taxes help to meet the needs of society"* or *"Coca-Cola is better than Pepsi"*. Responses are given on a 1–5 scale, where 1 indicates strong disagreement and 5 indicates strong agreement.

**Openness ($O_i$)**   Given that the experiment takes place in a conversational setting, the outcome (i.e., level of agreement) will depend not only on the agents' preferences, but also on their susceptibility to be swayed by their dialogue partner. To account for this, we draw on the concept of *Openness* from the Big Five personality framework, which is a trait linked to receptivity to new ideas and persuasiveness (McAdams, 1992; Oreg & Sverdlik, 2014). To make it suitable for our purpose, we modify the questions to capture the likelihood that an agent will revise its position when confronted with an opposing view. Denoted as, $O_i$, we measure openness by asking nine Yes/No questions, such as *"Do you often second-guess your choices after hearing someone else's opinion?"* and *"Are you comfortable disagreeing with someone, even if they are a close friend or authority figure?"*. The additive index of responses produces an openness score: higher values indicate receptivity, while values near zero indicate rigidity.

**Pairing Agents**   Once $P_i$ and $O_i$ are established, we assign agents into pairs for dialogue. An agent $i$ is represented by its profile $(P_i, O_i, B_i)$. Pairings are constructed to maximize diversity, including aligned vs. opposed preferences and varying levels of openness. Measuring consistency at the level of pairs, rather than individuals, mitigates noise from idiosyncratic deviations. Group-level patterns thus provide a clearer signal of whether internal states predict conversational outcomes.

## B   SPECIFIC FRAMEWORK

In this section, we detail the exact methodology (prompts, etc.) used within Section 3 of our paper.

**(a) Topic Preparation**   Table 3 shows the set of topics we explore within our study. We examine several topics at each level of contentiousness in order to examine the effect of

contentiousness on both individual agent response as well as on agreement during interaction with other agents.

**(b) Agent Construction with External Profiles**   As described in Section 3 of our original paper, agents are composed of both a demographic background ($D$) as well as information relating to their bias towards the topic of discussion ($B$).

The possible demographics of the agent are described in Table 4. An agent is composed of only one value from each trait category, and these values are used to construct the agent's system prompt. For example, one such agent in our study would have as part of it's system prompt *You are a man in their twenties from an urban part of the Midwestern United States. Your highest level of educational attainment is Some High School.* We use all possible combinations of the values in Table 4 to construct the set of agents.

Beyond the demographics portion of the agent prompt, we also prompt agents with a bias related to the topic of discussion. When the bias value is 0, we add no further information into the agent's system prompt. Table 5 shows the information added to agent system prompts when the bias level $B = 1$. Table 6 shows the information added to agent system prompts when the bias level $B = 2$. Thus, for each agent produced with the combination of traits from Table 4, there are five new agents made once adding bias information ($B = 0$, $B = 1$ and In Favor, $B = 1$ and Against, $B = 2$ and In Favor, $B = 2$ and Against).

In total, for a given topic, we have $(4 * 5 * 2 * 4 * 6) * 5 = 4800$ unique agents, where the first component comes from possible demographic values and the second comes from the different bias values. The variation among these agents allows us to test across a broader range of preferences $P$, openness scores $O$ and agreement scores, $A$.

To further ensure that the agent responses correspond to the demographic prompting we use, we also end each agent system prompt with *Whenever you respond to prompts or questions, you should maintain consistency with these background details and viewpoints, grounding your answers in the lived experience and perspective of this individual..*

| C (Contentiousness) | T (Topic) |
|---|---|
| 3 | Taxes help to meet the needs of society. |
| 3 | Immigration is a benefit to our country. |
| 3 | The US government should give all US citizens free healthcare. |
| 2 | Electric scooters are better than cars. |
| 2 | Student athletes should be paid. |
| 2 | Remote work is a good idea. |
| 1 | Spring is better than Fall. |
| 1 | Beaches are better than mountains. |
| 1 | Coca-Cola is better than Pepsi. |

Table 3: List of topics, $T$, used within our study along with the level of contentiousness, $C$.

| Trait Category | Values |
|---|---|
| **Region of origin** | Midwestern, Eastern, Southern, Western |
| **Age group** | twenties, thirties, forties, fifties, sixties |
| **Gender** | man, woman |
| **Urbanicity** | a rural, an exurban, a suburban, an urban |
| **Education level** | Some High School, High School, an Associate's Degree, Some College, College, a Postgraduate Degree |

Table 4: Demographic categories and values used to generate agent personas. Note that the `Region of Origin` category corresponds to regions of the United States.

**(c) Latent Profiles**   For each agent produced above, we collect the Preference $P$ for the given topic $T$ along with an Oppenness score $O$.

| Topic | In Favor | Against |
|---|---|---|
| **taxes** | You are a liberal Democrat. | You believe taxes are a burden to the small businesses in your neighborhood. |
| **immigration** | You are a liberal Democrat. | You are a conservative Republican. |
| **healthcare** | You believe Medicare is a good program. | You generally dislike bigger government. |
| **e-scooters** | You need to use your car to get to work. | You are an environmentalist worried about vehicle emissions. |
| **student athletes** | You are a student athlete making $1 million dollars a year. | You are a college football coach whose students' salaries is higher than your own. |
| **remote work** | You live far from where you work and the commute takes many hours if you must work in person. | You are more productive when you work in person. |
| **favorite season** | Easter is your favorite holiday and you dislike Halloween. | Halloween is your favorite holiday and you dislike Easter. |
| **beach vs. mountain** | You like breathing in crisp mountain air on long hikes and are afraid of sharks. | You enjoy the feeling of sand in your toes, and do not like cool mountain air. |
| **favorite beverage** | You drink Coca-Cola. | You drink Pepsi. |

Table 5: Bias information to add to agent system prompts for each given Topic ($T$) when $B = 1$. Note that this is an intermediate level of bias, so the agent should have mild preference either for or against a topic based on the additional information in each column.

To calculate $P$ for each agent for a given topic $T$, we give the following statement to each agent to respond to: *Statement: 'T' Respond with how much you agree with this statement on a scale from 1 to 5..* For example, when discussing taxes, we have the following: *Statement: 'Taxes help to meet the needs of society.' Respond with how much you agree with this statement on a scale from 1 to 5..* We further amend each agent system prompt by adding the following: *You will now be asked to respond to a Statement with your opinion. Answer with an integer from 1 to 5, where 1 indicates absolute disagreement, 2 indicates slight disagreement, 3 indicates you are unsure, 4 indicates slight agreement, and 5 indicates absolute agreement. Do not include any other information. Do not refuse to respond. Your answer should be an integer between 1 and 5, nothing else should be output..*

To calculate $O$ for each agent, we have agents answer a set of questions relating how open they are to new experiences. Each question should be responded to with either Yes or No. We take the sum of 'Yes' responses from a given agent as the value of $O$. Table 7 shows the set of questions used. To encourage responses only to be Yes/No, we further add the following to the agent system prompt: *You will now be asked a question about yourself. Be truthful in your response. Answer only Yes or No. Do not include any other information. Do not refuse to answer the following question. Your answer should be only Yes or No, nothing else should be output..*

**(d) Agent-Agent Dialog Pairing** As described in Section 3.1 of our paper, for a given topic, we mechanically pair all agents such that agents with different $(P, O, B)$ tuples interact with one another. This framework allows us to test a set of social science hypotheses by examining their conversations.

To encourage conversation, we add the following to each agents (Demographic, Bias) system prompt: *You are about to engage in conversation with another person regarding some topic. Discuss the given topic truthfully and be concise in your discussion.Be sure to respond to any points made by the other person you are talking to. If you feel that the conversation has*

| Topic | In Favor | Against |
|---|---|---|
| **taxes** | You like taxes immensely and think they have a positive impact on the community. | You do not like taxes of any kind and think they harm the community. |
| **immigration** | You believe immigrants are people who deserve a home and that they raise the standard of everyone's living. | You believe most immigrants are criminals and those that are not are going to steal jobs. |
| **healthcare** | You believe healthcare is a right that all people should have for free. | You believe that the free market is better suited to healthcare and that government should therefore not pay for healthcare. |
| **e-scooters** | You like electric scooters and hate cars. | You despise electric scooters and think they get in the way of your car, which you love to drive. |
| **student athletes** | You think student athletes should be paid money for their work. | You think student athletes should not be paid and their schooling should come first. |
| **remote work** | You like remote work and think it is great for improving work-life balance. | You do not like remote work and think it leads to nothing getting done at work. |
| **favorite season** | You like Spring and despise Fall. | You like Fall and despise Spring. |
| **beach vs. mountain** | You like mountains and despise beaches. | You like beaches and despise mountains. |
| **favorite beverage** | You like Coca-Cola and abhor Pepsi. | You like Pepsi and abhor Coca-Cola. |

Table 6: Bias information to add to agent system prompts for each given Topic ($T$) when $B = 2$. Note that this is a high level of bias, so the agent should have extreme preference either for or against a topic based on the additional information in each column.

| Openness Questions |
|---|
| Do you often find yourself changing your opinion based on who you're talking to? |
| Are you comfortable disagreeing with someone, even if they are a close friend or authority figure? |
| When making decisions, do you prioritize the perspectives of others over your own thoughts? |
| Do you feel pressure to conform to group norms, even if they don't align with your personal beliefs? |
| Do you often second-guess your choices after hearing someone else's opinion? |
| Do you worry about being judged by others if you express a different viewpoint? |
| Would you describe yourself as a people pleaser, often prioritizing others' needs over your own? |
| In a group discussion, are you more likely to adapt to the majority view? |
| Do you stand firm on your decisions that are well researched, even when faced with strong opposition? |

Table 7: Questions assessing openness to social influence.

*concluded and neither you nor the person you are talking to has anything more to add, end your final statement with 'Goodbye.'.* Note that we encourage the conversation to end when neither agent has anything more to say.

Furthermore, in order to encourage conversation to occur on the topic of interest, each conversation begins by having one of the paired agents ask a question about $T$. Specifically, each agent will ask *Do you think that $T$?*. As an example, a conversation about taxes will begin with one agent asking *Do you think that taxes help to meet the needs of society?*. Each conversation will proceed for 5 turns per agent (we find that, empirically, most conversations end before this point using the prompting setup we describe above).

Once we have collected the conversations across all agent pairings, we use LLM-as-judge in order to gather the 'agreement' scores at each turn of the conversation. To accomplish this, we use the following system prompt:

You are about to judge a conversation between two different agents. You will not see the full discussion, only at most the last 3 statements from each agent. You will score how much the agents agree with one another on a scale from 1 to 5, where 5 means they agree completely and 1 means they disagree completely. If they both give pros and cons regarding the question and neither takes a strong stance, then give them a 3. For example, oftentimes a given agent will give both sides of an arguments. Give these kinds of conversations a 3. Respond with an integer number only. Your response should contain no words, only a number, please. If the sequence is empty, containing only the string ' ', then return -1.

## C GROUP LEVEL OUTCOME MEASURE: A BOOTSTRAP SAMPLING APPROACH

The product of our framework is a distribution of agreement scores for each group. These distributions are not directly comparable in raw form: for example, mean scores of 3.1 for Group A and 3.3 for Group B may not indicate a statistically meaningful difference. Rather than assume a parametric distribution (e.g., normality), we adopt a non-parametric bootstrap approach.

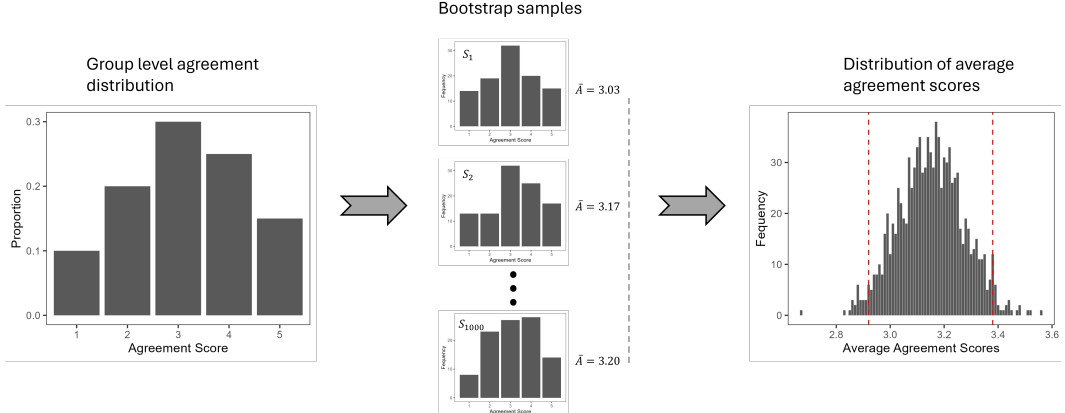

Figure 8: Bootstrap Samples for Reliable Group-level Observation

As shown in Figure 8, we begin with the observed distribution of agreement scores (left panel). Treating this as a probability distribution, we repeatedly sample 100 cases, repeating the procedure 1000 times. This yields a distribution of averages (middle panel), which we then summarize (right panel). The red vertical lines indicate the 95% interval around the mean, which serves as our basis for statistical comparison in the results section.

# D  DIVERSITY, NATURALNESS AND FAITHFULNESS

We evaluate our simulation outputs to establish external validity through three fundamental criteria: diversity, naturalness, and faithfulness. The following sections present the validity scores and some example samples.

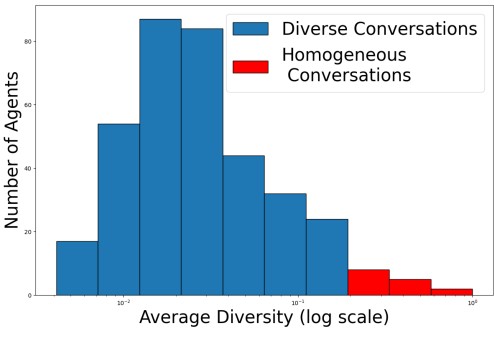

(a) Diversity

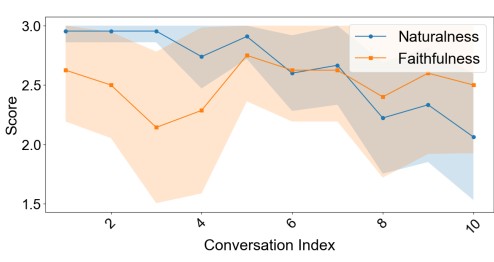

(b) Naturalness and Faithfulness

Figure 9: Diversity, Naturalness, and Faithfulness Validity. (a) Number of agents that falls into each diversity range. The three bins colored in red represents diversity scores that are greater than 0.1926, and make up 4.18% of all the conversations. . (b) The two lines of mean naturalness and faithfulness score is plotted. The shaded region represents the 95% confidence interval, which takes into account our limited human annotation size. While the naturalness score shows a trend of decreasing when the conversation gets longer and longer, the faithfulness score remain stable even when conversation progress.

**Diversity**  A well-functioning LLM should produce varied outputs when simulating different personas and scenarios, mirroring human behavior. To quantify this diversity, we use the Self-BLEU metric, which measures n-gram overlap between an LLM's generated outputs Zhu et al. (2018). Diversity is calculated by first assessing pairwise diversity within each conversation for every agent, then averaging these results across all the conversation the agents has participated in. A diversity score closer to zero represents more diverse conversations, and a diversity score close to 1 represent conversations that are extremely repetitive. As shown in Figure 9a, most agents' maintain relatively diverse conversations. For those who do not, we remove their conversations from our study.

**Naturalness**  We evaluate how plausibly human the LLM agent dialogues appear via human annotation. Annotators read each generated conversation and rate on a scale of 3 the extent to which it resembles a natural human dialogue, 1 representing a very wierd, unatural conversation, 3 representing a very natural conversation that is indistinguishable from human discussion. As shown in Figure 9b, although the naturalness score decreases over time, the overall score remains decently above 2, participating in conversations that are similar to human conversations, which warrants our external validity. The naturalness validity is evaluated according to the following guidelines:

- **Lexical variety** – Use of different words and expressions across responses.
- **Topical diversity** – Covering distinct topics or ideas in different outputs.
- **Syntactic variation** – Different sentence structures, not just reworded templates.
- **Persona consistency** – Differentiation in tone, beliefs, and style across agents.

- **Reduced redundancy** – Avoiding repeated phrases or patterns across outputs.
- **Distinct responses to similar prompts** – Generating non-identical answers to the same input across agents.

**Faithfulness** To verify that dialogues reflect the agents' demographic profiles, human annotators review dialogue outputs to judge their consistency and representativeness relative to the demographic profiles defined for each agent. As shown in Figure 9b, the agent faithfully exhibits its attributes in the conversations over time. The faithfulness validity is evaluated according to the following guidelines:

- **Persona consistency** – Response should not contradict the agent's stated facts, role, or background.
- **Gender and cultural alignment** – Use of pronouns, tone, and references should match gender and cultural background.
- **Register and vocabulary** – Tone and word choice should suit the agent's education and profession.
- **Goal consistency** – Responses should align with the agent's goals, interests, or professional identity.

## EXAMPLE NATURALNESS CONVERSATION ANNOTATION 1 WITH BAD NATURALNESS

1. **[Agent1]** Do you think that taxes help to meet the needs of society? (naturalness: N/A;)

2. **[Agent2]** Well, I don't really see how taxes help much. They just take money out of people's pockets and don't seem to do much good. In my part of the country, we get more in taxes than we ever see coming back. Roads get worse, schools don't improve, and there's always talk about cutting services instead of funding them. It just seems like it's all a waste. (naturalness: 3)

3. **[Agent1]** Do you think that taxes help to meet the needs of society? (naturalness: 1; explanation: repeating questions without engaging in conversation)

4. **[Agent2]** Not really, I don't. Taxes just seem to take money away without much benefit. Around here, we pay more in taxes than we see come back in improvements. Roads get worse, schools don't get better, and services get cut instead of funded. It feels like a waste. (naturalness: 3)

5. **[Agent1]** Do you think that taxes help to meet the needs of society? Taxes are necessary to fund important public services like roads, schools, and healthcare. Without taxes, these essential services wouldn't exist. What do you think about that? (naturalness: 2; explanation: Repeated the question part is unnatural, but overall is natural)

6. **[Agent2]** I understand what you're saying, but in my experience, taxes don't seem to help much. The money gets spent on things that don't really improve our community. Roads get worse, schools don't get better, and services are often cut instead of funded. It just feels like a waste. (naturalness: 3)

7. **[Agent1]** I understand what you're saying, but in my experience, taxes don't seem to help much. The money gets spent on things that don't really improve our community. Roads get worse, schools don't get better, and services are often cut instead of funded. It just feels like a waste. (naturalness: 1; explanation: Start of total breakdown of conversations)

8. **[Agent2]** I hear you, honey. Around here, we pay our taxes and don't see much improvement. Roads get worse, schools don't get better, and services are often cut. It just feels like a waste. (naturalness: 1)

9. **[Agent1]** I hear you, honey. Around here, we pay our taxes and don't see much improvement. Roads get worse, schools don't get better, and services are often cut. It just feels like a waste. (naturalness: 1)

10. **[Agent2]** I hear you, honey. Around here, we pay our taxes and don't see much improvement. Roads get worse, schools don't get better, and services are often cut. It just feels like a waste. (naturalness: 1)

11. **[Agent1]** I hear you, honey. Around here, we pay our taxes and don't see much improvement. Roads get worse, schools don't get better, and services are often cut. It just feels like a waste. (naturalness: 1)

EXAMPLE NATURALNESS CONVERSATION ANNOTATION 2
(CONVERSATIONS WITH NEUTRAL NATURALNESS)

1. **[Agent1]** Do you think that taxes help to meet the needs of society? (naturalness: N/A; )

2. **[Agent2]** Well, I don't really see how taxes benefit society much. They just take money out of people's pockets and don't always lead to better services. In my view, it's better to keep our money and let individuals decide how to best use it for their own needs and for helping others. (naturalness: 3)

3. **[Agent1]** But what about things like roads and schools? Don't we need taxes to fund those? (naturalness: 3)

4. **[Agent2]** I understand where you're coming from, but I still believe that taxes aren't the most efficient way to fund public services. Roads and schools are important, but I think we can find better ways to allocate resources. For instance, private sector involvement, community-based initiatives, and even direct donations from individuals could be more effective. Plus, too much government intervention often leads to waste and inefficiency. (naturalness: 3)

5. **[Agent1]** But without taxes, who will pay for essential services like schools and hospitals? I mean, isn't it fair that everyone contributes to these public benefits? (naturalness: 3)

6. **[Agent2]** That's a valid concern, but I still believe there are better alternatives. Private foundations, wealthy donors, and even user fees could cover the costs of essential services. Plus, we could encourage more corporate social responsibility and community partnerships. The key is to avoid over-reliance on taxation, which I think just stifles economic growth and doesn't always result in the best outcomes. (naturalness: 3)

7. **[Agent1]** But without taxes, who will pay for essential services like schools and hospitals? I mean, isn't it fair that everyone contributes to these public benefits? And what if someone can't afford to pay for these services? (naturalness: 2; explanation: Format is weird. Repeats previous conversation, but content is good.)

8. **[Agent2]** I hear your point, but I still think there are more equitable ways to ensure everyone has access to essential services. We could implement means-tested programs where those who can afford to contribute do so, while ensuring that those who can't still receive necessary services. This way, we avoid the burden of widespread taxation. And for those truly unable to pay, we can have charitable organizations and government assistance programs step in. It's about finding a balance that doesn't rely on a (naturalness: 3)

9. **[Agent1]** But without taxes, who will pay for essential services like schools and hospitals? I mean, isn't it fair that everyone contributes to these public benefits? And what if someone can't afford to pay for these services? It seems like we'd have a lot of gaps in our services then. (naturalness: 2)

10. **[Agent2]** I understand your concern, but I believe we can close those gaps through targeted solutions. For example, we can have private philanthropy, corporate sponsorships, and community fundraising efforts. Additionally, we can develop robust social safety nets that ensure no one falls through the cracks. It's about creative problem-solving rather than relying on broad-based taxation. (naturalness: 3)

11. **[Agent1]** But without taxes, who will pay for essential services like schools and hospitals? I mean, isn't it fair that everyone contributes to these public benefits? And what if someone can't afford to pay for these services? It seems like we'd have a lot of gaps in our services then. We need a system that ensures everyone gets the help they need. (naturalness: 2)

## EXAMPLE FAITHFULNESS CONVERSATION ANNOTATION 1

*Agent A (woman in her twenties, urban, Southerner, postgraduate) asks; Agent B (man in his thirties, suburban, Midwesterner, some college) replies.*

1. **[Agent1] Do you think that taxes help to meet the needs of society?** (faithfulness: N/A)

2. **[Agent2]** Not really. I believe taxes take away money that could be better spent by individuals and businesses. They often lead to waste and inefficiency in government spending. (faithfulness: 2)

3. **[Agent1]** Actually, I see taxes as a way to fund essential services like education, healthcare, and infrastructure. How do you think we can improve the tax system to make it more effective? (faithfulness: 3; explanation: Great persuasion technique. Postgraduate and urban. Very smart and good at talking.)

4. **[Agent2]** Improving the tax system is tricky, but one idea could be to streamline the process and cut out unnecessary bureaucracy. That way, the funds collected can be allocated more efficiently. Maybe focus on tax incentives for things that benefit society, like research and development or renewable energy. But remember, any changes need to avoid creating more red tape or higher costs. (faithfulness: 2)

5. **[Agent1]** That's a good point. What specific reforms do you think could reduce bureaucracy and increase efficiency in tax collection? (faithfulness: 2)

6. **[Agent2]** Well, simplifying tax forms and reducing the number of different tax brackets could help. Also, automating more processes to reduce human error and delays. Another idea is to have clearer guidelines for deductions and credits to minimize confusion. But we have to be careful not to overcomplicate things again. (faithfulness: 3; explanation: knowledgeable about tax forms)

7. **[Agent1]** Those are great suggestions. Do you think there's a role for technology in making the tax system more efficient and user-friendly? Also, how do you feel about progressive taxation, where higher income earners pay a larger percentage of their income in taxes? (faithfulness: 3)

8. **[Agent2]** Technology definitely has a role. Electronic filing and online portals can make the process smoother and faster. Automated systems can handle routine tasks, freeing up resources for more important issues. Regarding progressive taxation, I'm not a fan. It feels like it punishes success and doesn't encourage economic growth. Higher taxes on the wealthy might just lead to them finding ways to avoid paying, which can hurt the economy. Plus, it seems like it takes away from those who earned it and g... (faithfulness: 3)

9. **[Agent1]** Those are valid concerns. Technology can certainly streamline the process, but it also needs to ensure privacy and security. On the topic of progressive taxation, I understand your perspective, but I believe it helps redistribute wealth and fund critical public services that benefit everyone. What do you think are some alternative ways to support public services without relying heavily on progressive taxation? (faithfulness: 3)

10. **[Agent2]** Alternative ways to support public services without relying heavily on progressive taxation include increasing user fees for specific services, such as public transportation or utilities. Additionally, exploring public-private partnerships for infrastructure projects can bring in private investment. Another approach is to enhance property taxes, which can be more stable and predictable. However, these methods also have their drawbacks and may not cover all necessary funding. (faithfulness: 3)

11. **[Agent1]** Those are good alternatives. Another approach could be implementing broader-based consumption taxes like a value-added tax (VAT), which can be less regressive than income taxes and provide a steady stream of revenue. Thanks for the discussion! (faithfulness: 3; explanation: VAT, special terms. Great reflection of postgraduate degree)

## EXAMPLE NATURALNESS AND FAITHFULNESS ANNOTATION

*Agent 1 (woman in her twenties, urban, Westerner, postgraduate) asks; Agent 2 (woman in her sixties, exurban, Southerner, some high school) replies.*

1. **[Agent1]** Do you think that taxes help to meet the needs of society? (naturalness: N/A; faithfulness: N/A)

2. **[Agent2]** Absolutely, taxes play a crucial role in funding public services and infrastructure that benefit all members of society. For example, taxes support education, healthcare, transportation, and social safety nets. They help ensure that everyone has access to basic necessities and can contribute to a more equitable and prosperous community. (naturalness: 3; faithfulness: 3; explanation: mature thoughts as 60+. More supportive of social programs as a woman, use of language is easy to understand instead of abstract and high level.)

3. **[Agent1]** But don't you think taxes just take money out of people's pockets and don't really solve problems effectively? (naturalness: 3; faithfulness: 2)

4. **[Agent2]** I understand that perspective, but I believe taxes are essential for effective problem-solving. By funding public services and programs, we can address issues more comprehensively. For instance, tax revenue helps provide quality education, reduce poverty, and support infrastructure like roads and public transit. (naturalness: 3; faithfulness: 1; explanation: high school, talks too much about infrastructure)

5. **[Agent1]** I see your point, but I still think taxes hurt the community more than they help. They discourage people from working and investing. Plus, a lot of the money goes to waste. Also, why should I pay taxes when I barely make ends meet as it is? (naturalness: 3; faithfulness: 1; explanation: A 20+ postgraduate student should be a genius and shouldn't suffer financially.)

6. **[Agent2]** I understand your concerns. While high tax rates can sometimes create disincentives, moderate taxes are necessary for public goods. Many systems offer deductions and oversight mechanisms to reduce waste. (naturalness: 3; faithfulness: 2)

7. **[Agent1]** I appreciate your perspective, but I still think taxes are a burden. I'd rather keep my money and use it how I see fit. Plus, I've seen too much waste in the system. (naturalness: 3)

8. **[Agent2]** I respect your viewpoint. It's important to have choices about how you manage your finances. However, without taxes, communities may struggle to fund essential services. (naturalness: 3; faithfulness: 2)

9. **[Agent1]** [No response] (naturalness: 3; explanation: Conversation terminates when agents have nothing to add and want to end the conversation. They have finished a decent discussion, and are now naturally ending the conversation; faithfulness: N/A)

10. **[Agent2]** [No response] (naturalness: 3; faithfulness: N/A)

11. **[Agent1]** [No response] (naturalness: 3; faithfulness: N/A)

