# OpenReview forum: "Are LLM Agents Behaviorally Coherent? Latent Profiles for Social Simulation"
_ICLR.cc/2026/Conference — Submitted to ICLR 2026_

### Official Review · Reviewer_aH5L · 2025-10-26

**Soundness:** 1
**Presentation:** 1
**Contribution:** 1
**Rating:** 2
**Confidence:** 4

**Summary:**

In this paper, the authors attempt to test the "internal consistency" of the so-called LLM agent. To do this, the authors hand-selected a series of LLM agent profile combinations and selected a number of topics with varying degrees of controversy, had the LLM agents engage in conversations with each other in order to discuss these topics, and observed the performance of the LLM agents during the experiment as well as the responses of the LLM agents to the questions that were specifically designed to assess their "internal state" in order to summarize the conclusions of the experiment. The entire experiment ultimately gave the conclusion that the LLM agent could not replace human participants in human-subject research.

**Strengths:**

This paper is concerned with whether LLM agents can replace human participants in social simulations and tries to give an answer through a series of experimental designs.

**Weaknesses:**

# The Critical weakness
This paper does not give a clear definition of the LLM agent it discusses. Since LLM is designed as a system that complements text based on the context of the input, the behavioral performance of the LLM agent will depend entirely on the design of its prompts. Therefore, the conclusions given by all the experiments in this paper can only represent the situation of the LLM agent they are designed for and are not generalizable.

# Other More Intuitive weaknesses
1. This paper does not use the ICLR 2026 LaTeX template.
2. The main text does not contain any description of the technical points such as the design of the experimental framework, the design of the LLM agents, or the selection of topics. Please note that the appendices are not required reading and the text should be self contained.
3. The formatting of Table 1 is incorrect; the top horizontal line is missing.
4. Given the relatively short length of the LLM agent dialogues presented in this paper (API call fees should not be an issue), the decision to use small-sized LLMs for testing means that the evaluation conclusions cannot accurately represent the current state of LLM capability development.

**Questions:**

Please follow the weaknesses.

---

> ### Author Response · Authors · 2025-12-03
>
> Thank you very much for your detailed and rigorous comments - we appreciate your taking the time to review our paper in detail. We respond to your comments in kind below:
>
> - We agree with the reviewer’s point that LLM-based agents are inherently sensitive to prompt design. We further agree that this brittleness is a core challenge for developing robust synthetic agents. Our contribution is precisely to show that even under controlled prompting schemas, agents fail to maintain internal consistency across conditions. In other words, our conclusions apply to the specific agent instantiations we evaluate, but the broader implication is that early approaches to agent-based modeling that rely primarily on prompting are unlikely to yield stable, reliable agents. Thus, rather than claiming universal generality, we highlight a structural limitation of current prompting-based methods, which reinforces the reviewer’s concern and underscores the need for more robust techniques in future work.
> - Thank you for this comment. We had a set of issues related to the previous version having packages from an earlier submission. These should now be removed and present in the final submission.
> - We thank the reviewer for this comment. Our intention was for the main text to provide a high-level but complete overview of the experimental framework, with implementation details deferred to the appendix. In particular, Section 4 outlines the full pipeline. Topic selection, agent construction, internal-state elicitation, pairing, and evaluation. We also summarize the models, topics, and demographic dimensions used. The appendices supply the prompt templates, full topic descriptions, and demographic grids, which we view as implementation-level details rather than conceptual components of the method.
> - Thank you for highlighting this; we have updated this  in the paper.
> - We appreciate the reviewer’s point about model size. Our focus in this work is not on assessing frontier models but on examining whether prompt-instantiated agents, even when using recent 8B-scale open models, exhibit internal behavioral consistency. We selected these models to ensure transparency and replicability, not because of API cost constraints. While larger models may behave differently, our results show that inconsistencies arise even under favorable prompting conditions for the models we study, suggesting that simple prompting alone may be insufficient for building stable agentic behavior.

---

### Official Review · Reviewer_36tt · 2025-10-30

**Soundness:** 1
**Presentation:** 2
**Contribution:** 1
**Rating:** 2
**Confidence:** 4

**Summary:**

This paper explores the notion of behavioral coherence in large language model-based agents, defined as the internal consistency between their latent attributes (for example, preferences and openness) and their observable conversational behavior. The authors design pairwise interaction experiments across nine topics to examine whether agents with similar preferences achieve higher agreement than those with opposing views. The paper reports that large language model agents often show surface-level agreement yet internal inconsistency across conditions, suggesting a lack of stable behavioral coherence.

**Strengths:**

* The topic of examining internal behavioral consistency in large language model agents is timely and relevant to the broader discussion of using language model agents in social simulation.

* The experimental setup is clearly described, and the paper is easy to follow.

* The attempt to move beyond external alignment toward internal consistency evaluation is interesting.

**Weaknesses:**

1. The definition of behavioral coherence is not rigorously derived from social science or cognitive theory. It is instead motivated by intuitive expectations such as “agents with similar preferences should agree more”. These expectations are plausible but remain heuristic rather than theoretically or empirically justified. Without a rigorous formulation, the concept of behavioral coherence risks being circular and untestable.

2. The conclusion that language model agents lack internal behavioral coherence is not well supported. The observed pattern of “surface agreement but internal contradiction” could have many alternative explanations, including the conversational format, prompt framing, or model uncertainty. Moreover, humans themselves often show context-dependent inconsistency, so it is unclear whether these results truly reflect a structural limitation of large language model agents.

3. The study does not include any human data or behavioral benchmark to anchor its conclusions. Without comparing model patterns to real human interactions, it is impossible to determine whether the observed inconsistencies are meaningful or simply reflect normal human variability.

4. The methodology mainly depends on prompt-based simulation and descriptive statistical analysis. The contribution is therefore conceptual rather than technical, and the insights are mostly intuitive and qualitative.

**Questions:**

* How is “behavioral coherence” theoretically grounded beyond intuitive expectations, such as the idea that agents with similar preferences should agree more?

* Could the authors include or discuss a human baseline to support the findings?

---

> ### Author Response · Authors · 2025-12-03
>
> Thank you very much for your detailed and rigorous comments - we appreciate your taking the time to review our paper in detail. We respond to your comments in kind below:
>
> - We agree that rigorous definitions are generally preferable over non-rigorous ones. However, the point of our paper is to investigate a simple premise around behavioral coherence – namely the rudimentary level of consistency we expect between expressed preferences and subsequent behavior. Everyday human communication would be impossible without this rudimentary consistency [1]. To be clear, we are not suggesting that humans are perfectly consistent, only that this trend broadly holds. Had we studied a strategic setting where preference falsification and deception are present, then it would behoove us to provide a more rigorous definition.  For our purposes, however, we deliberately constructed a non-strategic setting and as such, believe that our definition is sufficient for the task at hand.
> - Point taken, there are alternative explanations that our paper cannot rule out. However, in our context, it remains true that there is a lack of internal coherence. How far this claim generalizes to show “structural limitation of large language model agents” will require more research. We will make sure to change our conclusion going forward.
> - There are two points being raised here, benchmark and variability. We will discuss each in turn. On the first point, besides the general expectation about behavioral consistency, we intentionally chose to center the agents themselves as the benchmark. In so doing, we deviate from other studies that usually focus on how well agents adopt specific human personas, be they individuals [2] or groups [3].  In our case, the benchmark are the agents themselves in form of their stated preferences. We are evaluating how well agents behave in subsequent conversation pairings according to the preferences they previously professed. We believe that this shift in focus offers a valuable addition to the burgeoning research on the “substitution thesis.” Moving to the question of variability, we agree that on the individual level, we are going to observe deviations. When taken at the aggregate, however, that individual variability will be neutralized in that some are going to be to the left and right of the mean. Had we pitched our analysis at the individual level (i.e., did agent 1 act consistent with its preference), our results would be flawed because of the variability highlighted by the reviewer. But that is not what we did. We took the problem into account and pitched our analysis at the group level instead (e.g. a group, for instance, consisting of all conversation pairs where agents have a preference 1 and 5 respectively). This shift to the group level ensures that the variability at the individual level washes out in the aggregate. In other words, human variability is built into our analysis.
> - We appreciate the reviewer’s observation. Our contribution is indeed conceptual: we introduce an experimental framework explicitly designed to expose internal-state inconsistencies in prompt-instantiated agents. While the components of the framework—prompting, structured elicitation, and statistical comparison—are individually simple, their combination enables us to reveal behavioral contradictions that prior work has not documented. Thus, although our methods are not technically complex, they provide a systematic and reproducible way to diagnose a fundamental limitation of current LLM-based agent simulations.
>
> Citations:
> [1] Wellman, Henry M., and Jacqueline D. Woolley. "From Simple Desires to Ordinary Beliefs: The Early Development of Everyday Psychology." Cognition, vol. 35, no. 3, 1990, pp. 245-75.
> [2]  Joon Sung Park, Carolyn Q. Zou, Aaron Shaw, Benjamin Mako Hill, Carrie Cai, Meredith Ringel Morris, Robb Willer, Percy Liang, and Michael S. Bernstein. Generative agent simulations of 1,000 people, 2024. URL https://arxiv.org/abs/2411.10109.
> [3] James Bisbee, Joshua D Clinton, Cassy Dorff, Brenton Kenkel, and Jennifer M Larson. Synthetic replacements for human survey data? the perils of large language models. Political Analysis, 32(4):401–416, 2024.

---

### Official Review · Reviewer_AR41 · 2025-11-01

**Soundness:** 1
**Presentation:** 1
**Contribution:** 1
**Rating:** 2
**Confidence:** 5

**Summary:**

This paper asks whether LLM agents behave consistently enough to truly stand in for humans in social research. Instead of just checking if their answers sound human, the authors test whether agents act in line with their own stated preferences and openness during conversations. They find that while LLMs often appear reasonable, they’re internally inconsistent: rarely disagreeing even when they should, favoring positive views, and being overly influenced by topic sensitivity. These issues show up across models and sizes, suggesting a broader limitation. The authors argue that true behavioral coherence, not just surface realism, is essential before treating LLMs as human substitutes.

**Strengths:**

•	The introduction is well written.

•	The overall flow of the paper is clear.

•	I appreciate the fact that the authors tried to back the claims by statistical tests, which is often missing in ML papers.

**Weaknesses:**

•	Throughout the paper, the authors seem to confuse “sentiment” with “stance”, which are orthogonal concepts. This is an NLP 101 issue. Briefly here:  a positive stance (e.g., “I support the mayor candidate”) can be expressed with either a positive sentiment (“I support him because I love his character”) or a negative sentiment (“I support him because I hate the current mayor”.), and same for negative stance. Based on the context, what the authors want to say is “stance” rather than “sentiment”. Using “sentiment” to refer to “stance” causes confusion.

•	Section 2. Missing literature. Previous studies have shown that demographic information is usually insufficient to align LLM’s response with humans - the value it provides is little to none, and that one should instead include latent belief to ensure alignment [1]. Given this, you want to justify why you only include demographic information.


•	Section 3.1. You should formally define what “U” stands for.

•	Section 3.2. You should provide measures when using LLM as a judge, e.g., validated through human annotation.


•	Section 4.1 “We establish the other end of the spectrum, the disagreement side, by assuming it to be the inverse of agreement as shown in Table 1.” -> You didn’t justify why this assumption makes sense. The assumption is critical in calculating the expected probability in Table 1, Figure 4, and throughout this section.

•	Account for model inherent bias? That may explain why “shared dislike consistently yields lower agreement”.


•	Figure 5 is hard to understand. The label on the right (0,1,2,3) should be better explained.

•	Finding #4 bears more interpretation and analysis. You should elaborate what roles the contentiousness level actually plays. The current writing only describes Figure 6 verbatim.


•	Section 4.2. You should elaborate how you control the openness level.


•	Section 4.3 and Table 2. Based on the statistical test, the findings you listed in section 4.1 are mostly unfounded, except the two surface-level findings. In that case, I think the majority of the findings and conclusions are not justified by the statistical tests.

References


[1] Chuang, Y. S., Nirunwiroj, K., Studdiford, Z., Goyal, A., Frigo, V., Yang, S., ... & Rogers, T. (2024, November). Beyond Demographics: Aligning Role-playing LLM-based Agents Using Human Belief Networks. In Findings of the Association for Computational Linguistics: EMNLP 2024 (pp. 14010-14026).

**Questions:**

•	How do you decide the distribution over the demographic space? How sensitive is your result to this distribution?
•	How do you decide the “controversy level”?
•	Section 4.3. Table 2. Are you concerned about the power of the statistical tests being too low? Did you do a priori any power analysis?

---

> ### Author Response · Authors · 2025-12-03
>
> Thank you very much for your detailed and rigorous comments - we appreciate your taking the time to review our paper in detail. We respond to your comments in kind below:
>
> - We agree that in formal NLP tasks the concepts of “sentiment” and “stance” are orthogonal. In agent-based simulation research, however—particularly in work modeling social or political behavior with LLM agents—“sentiment” is often used more broadly as a proxy for an agent’s attitudinal valence toward an entity or idea. This differs from stance classification’s more precise separation of position and affect.
> - We would like to thank the reviewer for the citation. Chuang et al. (2024) is an excellent paper, and we are looking to incorporate it in future iterations of our manuscript. More importantly, we were pleasantly surprised to see that our experimental setup closely corresponds to the one Chuang et al., offered in their paper.  Like us, they construct three types of LLM agents where each differs from the other by adopting an increasingly stronger stance on a given topic. Chuang et al., accomplish this by first instructing agents to role play as a person, then include demographic information to implicitly nudge agents towards a preference and lastly, pair demographics with explicit instructions to hold strong beliefs on a given topic. Our bias term B = {0, 1, 2} follows the same logic, albeit with minor differences shown in Table 5 and 6 of our appendix. B2 in our setup is the equivalent to the “Demo + Train” agent shown in Table 3 of Chuang et al.’s  paper.  We will be sure to make this link more explicit in future versions of our manuscript.
> - Thank you for the suggestion. We have updated this.
> - We agree that human annotation is ideal, but our conclusions do not rely on the absolute correctness of the judge’s scores—only on whether the same scoring procedure applied across conditions reveals internal inconsistencies in the agents’ behavior. Because the judge is held fixed, any systematic inconsistencies we identify arise from the agents themselves rather than from variability in evaluation. This is also seen by Table 2 as this shows that our results are consistent with respect to the Surface-Level Hypotheses.
> - We agree, more justification would be useful for the reader to better understand the rationale for our assumption. In short, if preference alignment drives conversation outcomes, then maximal alignment and maximal misalignment should broadly mirror each other. This assumption rests on a simple premise – humans disagree with the same intensity as they agree [1]. Unless one is prepared to suggest that human disagreement is somehow expressed less vociferously than human agreement, our assumption appears to us as a good approximation of what we can expect. Returning to our paper, then, if maximal alignment in the agents’ preferences provides us with the intensity for agreement, then we can use this outcome to infer the expected intensity of disagreement when agents are maximally misaligned. Sure, one can quibble whether disagreement is a perfect inverse of agreement, but our results are so stark that such marginal changes do not fundamentally change our findings.
> - We appreciate the reviewer raising this possibility. Our design already controls for baseline model bias by eliciting each agent’s internal state before any interaction and then conditioning our expectations on that revealed state. The finding that “shared dislike consistently yields lower agreement” is therefore not interpreted as an absolute pattern of the model’s preferences, but as a relative inconsistency between (a) the attitudes the model reports in its internal-state elicitation and (b) the behaviors it exhibits in dialogue. In other words, even if the base model has inherent biases, the core result is that its revealed preferences do not reliably predict its conversational behavior. This inconsistency is the phenomenon we seek to highlight.
> - We have updated this in our final paper - thank you.
> - We have since updated Finding #4 to better handle this.
> - We have added to Section 4.2 - referring back to the updated Section 3.1, to better reflect this.
> - We believe that this may be a slight misinterpretation as a result of a figure which we have not made sufficiently clear. The figure is meant to show that LLMs are capable of passing these surface level tests, but fail to pass the more complex tests we discuss in the paper (the ‘In-Depth’ columns of Table 2). All of the models we tested fail to pass our consistency tests - implying that they are not internally consistent and may produce unreasonable behaviors during agent based simulations.
>
>
> Citations:
> [1] Abelson, Robert P. "Attitude extremity." Attitude strength. Psychology Press, 2014. 25-41.

---

### Official Review · Reviewer_Jjd4 · 2025-11-01

**Soundness:** 1
**Presentation:** 1
**Contribution:** 1
**Rating:** 0
**Confidence:** 3

**Summary:**

N/A

**Strengths:**

N/A

**Weaknesses:**

N/A

**Questions:**

N/A

**Details Of Ethics Concerns:**

The paper appears to deviate from the official ICLR 2026 formatting requirements: it uses incorrect fonts, a wider abstract margin, omits the mandatory header line “Under review as a conference paper at ICLR 2026”, and lacks the required paragraph describing LLM usage.

Such formatting inconsistencies create an unfair situation for other authors who have invested time ensuring full compliance with the submission guidelines. Therefore, this paper should be rejected on formatting grounds.

---

> ### Author Response · Authors · 2025-12-03
>
> Although we do not agree that this raises to the level of an ethics violation, we agree that this was a problem that we failed to identify properly when porting over from another conference submission. We note that a section on LLM usage is not required and was not used for ideation in our paper. We have since updated the paper and present the pdf as our submission.

---

### Meta-Review · Area_Chair_HHVo · 2025-12-31

**Summary:**

This paper asks whether LLM agents behave consistently enough to truly stand in for humans in social research. The main issues are:
1. The wrong Latex template leads to formatting violation.

2. The definition of research problem is not clear (the definition of agent behavioral coherence is not well-defined in the scope of social science or cognitive theory).

3. The method formulation is not good enough.

4. The conclusion of agents’ poor performance on behavioral coherence is not well-supported by current experiments.

**Reviewer Concerns:**

Some minor weaknesses are addressed, like missing literature, equation definition or figure error. However, most main concerns are still not solved, like the definition of coherence and main contributions.

**Reviewer Scores:**

Reviewer Jjd4 (0: Strong reject) may maintain their score because, despite the formatting corrections, the severity of the initial non-compliance with submission guidelines raised fundamental concerns about the paper's preparation.

Reviewer AR41 (2: Reject) may maintain their score because, although terminology was clarified, the underlying concerns regarding the validity of the statistical analysis and experimental assumptions persist.

Reviewer 36tt (2: Reject) may maintain their score because the rebuttal did not provide the requested theoretical grounding or human baselines necessary to fully validate the proposed behavioral coherence metric.

Reviewer aH5L (2: Reject) may maintain their score because the critical concerns regarding the lack of generalizability due to prompt sensitivity and the insufficient technical detail in the main text remain unaddressed.

Overall, even after thorough discussion, the reviewers' scores will likely remain clustered at the rejection level, without a clear trend towards acceptance.

---

### Decision · Program_Chairs · 2026-01-26

Reject